# Four-Year Follow-Up of the Maternal Immunological, Virological and Clinical Settings of a 36-Year-Old Woman Experiencing Primary Cytomegalovirus Infection Leading to Intrauterine Infection

**DOI:** 10.3390/v15010112

**Published:** 2022-12-30

**Authors:** Gabriella Forner, Alda Saldan, Carlo Mengoli, Sara Pizzi, Marny Fedrigo, Nadia Gussetti, Silvia Visentin, Annalisa Angelini, Erich Cosmi, Luisa Barzon, Davide Antonio Abate

**Affiliations:** 1Department of Molecular Medicine, University of Padova, 35121 Padua, Italy; 2Pathology Unit, Padova General Hospital, 35121 Padua, Italy; 3Department of Cardiothoracic and Vascular Science and Public Health, University of Padova, 35121 Padua, Italy; 4Reference Center for Congenital Infections, Padova General Hospital, 35121 Padua, Italy; 5Department of Women’s and Child’s Health, University of Padova, 35121 Padua, Italy

**Keywords:** CMV cell-mediated immunity, congenital CMV

## Abstract

The present study aims to provide the sequential immunological, clinical and virological events occurring in a CMV-infected pregnant woman experiencing intrauterine CMV transmission. In brief, a case of primary CMV infection occurred in a 36-year-old pregnant woman. The patient exhibited early-sustained viremia and viruria, detectable presence of CMV in saliva concomitant with a strong CMV-specific cell-mediated response (427 EliSpots). CMV was detected in the amniotic fluid at 15 weeks of pregnancy (>1 × 10^6^ CMV copies/mL). The pregnancy was deliberately interrupted at 16 weeks of gestation. Fetal histological and pathological examinations revealed placentitis and fetal brain alterations as microcephaly and cortical dysplasia. Interestingly, this clinical report shows: (1) there was a rapid and sustained CMV-specific cell mediated immune response (Th1) in association with low IgG avidity (Th2) correlated with fetal CMV transmission. (2) The levels of CMV-specific cell-mediated immune response persisted at high levels up to 200 weeks after infection despite clinical and viral clearance. (3) The histological and pathological evidence suggests that a potent pro-inflammatory condition at the placental level may lead to cCMV.

## 1. Introduction

Human cytomegalovirus (CMV) represents one of most common causes of congenital infection in developed countries [1,2]. Primary CMV infection during the first trimester of gestation may be associated with fetal transmission with a rate ranging from 30% to 40%, suggesting that the host immune response or viral strain variability may influence the rate cCMV transmission [3,4]. Primary CMV infection is often associated with in utero fetal death, neurological sequelae, and development abnormalities. A higher prevalence of fetal damage occurs if the CMV infection is acquired during the first trimester of pregnancy as compared to the second and third trimester [1]. Infected children may exhibit severe neurologic and sensorineural disorders at birth or post-natal sequelae within the first years of life [5]. Fetal infection following maternal CMV infection within the first trimester is considered established when CMV-DNA is detected in amniotic fluid [6,7]. In our prior study, a non-invasive procedure, the CMV EliSpot, performed on maternal peripheral blood isolated cells, revealed that elevated levels of cell-mediated immunity (CMI) specific for CMV, significantly correlated with a higher risk of fetal transmission, thus suggesting that an imbalanced or altered pattern of Th1/Th2 response may be the ultimate cause of cCMV [8].

This case report provides a four-year follow-up of a cCMV case resulting from maternal primary CMV infection. During the follow-up study, a complete and comprehensive virological, immunological and pathological pattern was determined.

## 2. Material and Methods

### 2.1. Direct and Indirect Diagnostic Procedures and HIG Therapy

Primary maternal CMV infection was defined by the presence of specific IgG, IgM and low CMV specific IgG avidity (<25%). CMV avidity, IgM and IgG serology was performed on Liaison instrumentation (DiaSorin, Saluggia, Italy). CMV IgG avidity was calculated as percentage of un-dissociated CMV-specific IgG/total CMV-specific IgG. CMV-DNA was detected in blood and urine by quantitative real-time polymerase chain reaction (qPCR) as described previously [8]. Saliva swabs were inoculated in cell culture for viral isolation as described [9]. The virus was not isolated from the saliva swabs. Imaging analysis included fetal MRI and ultrasounds scans. Fetal pathological and histopathological examinations were performed at the University of Padova pathology unit. The ethical committee of the Padua General Hospital approved the study. Interferon-gamma (IFN-γ) EliSpot assays were performed as previously shown [10,11,12]. CMV-EliSpot data are shown as number of spots forming colonies (SFC)/2 × 10^5^ PBMCs) [10].

### 2.2. Case Report

In 2013 a 36-year-old Caucasian woman in her first pregnancy was referred to the maternal fetal Infectious Diseases Unit—Padua General Hospital, Italy—for prenatal evaluation concerning a possible active CMV infection. The anamnesis revealed that the pregnancy was conceived after an assisted reproductive procedure, involving in vitro fertilization. No symptoms were reported during the admission and during weeks before.

Serological screening performed at 5 weeks of gestation detected the presence of anti-CMV IgM and IgG and low CMV-specific IgG avidity (5%). CMV-DNA was detected in blood (>2 × 10^3^ copies/mL), urine (>5 × 10^3^ copies/mL) (Figure 1) and there was a positive detection of CMV in the saliva. Viremia and viruria persisted stably up to 12 and 27 weeks after infection, respectively. The blood cell counts revealed 42% of neutrophils and 47% of lymphocytes. All these results were consistent with a primary CMV infection occurring within the first weeks of pregnancy. IgG avidity values revealed a progressive maturation (5% to 88%) of the humoral immune response. The highest value (88%) was reached four years after the primary infection. A vigorous cell-mediated immune response (427 IFN-γ EliSpots) was detected at 5 weeks of gestation. The cell-mediated immune response reached its peak 21 weeks after the primary CMV infection (551 IFN-γ EliSpots) while the nadir point was detected at 32 weeks post-infection (265 IFN-γ EliSpots). EliSpot values remained constantly high at 182 and 200 weeks after the primary infection (383 and 344 IFN-γ EliSpots, respectively). The EliSpot positive controls (PMA + Ionomycin and phytoemoagglutinin) were consistently high (>1000 SFC/2 × 10^5^ PBMCs) throughout the study, suggesting that general cellular immune response was functional. The EliSpot and laboratory data prompted us to perform a rapid amniocentesis procedure. At 15 weeks of gestation, CMV was detected in the amniotic fluid with qPCR levels at 1 × 10^6^ copies/mL. The shell vial test confirmed the CMV positivity in the amniotic fluid. The fetal karyotype had no abnormalities. The ultrasound scans revealed fetal brain abnormalities compatible with cCMV infection, including microcephaly, bilateral ventriculomegaly with hyperecoic blood vessels and cerebral parenchyma subversion. Fetal MRI revealed cortical dysplasia and migration abnormalities with thickened and irregular germinal matrix (Figure 2A,B). After specialist counseling, the couple opted for termination of pregnancy and fetal post-mortem pathological examination was performed. Fetal histopathological examination disclosed cytopathic alterations characterized by viral inclusions in parenchymal and endothelial cells of the liver, lungs, pancreas, spleen, brain (basal plate), ovary, thyroid gland and thymus. Brain histological examination revealed lateral ventriculomegaly and reduced migration of germinal cells towards the cortex. Placental histological examination evidenced stromal fibrosis and a decreased number of stromal blood vessels, mineralization and deposition of hemosiderin on the basal membrane (evidenced by positive Perls staining) and a low grade of lymphocytic and plasma cellular infiltrate of villous stroma (Figure 3A,B). Microscopic evaluation evidenced cytomegalic stromal cells with eosinophil intranuclear inclusion bodies and clusters of intracytoplasmatic basophile inclusion bodies (Figure 3C). CMV immunostaining evidenced strong and diffuse nuclear positivity (Figure 3D). Furthermore, hypoxia-associated alterations were expressed by microfoci of stromal necrosis and rare calcification, increased syncytial knotting (evidenced by the Tenney–Parker changes) and perivillous fibrin deposition. The morphological and immunophenotypic analysis was therefore suggestive of CMV placentitis.

## 3. Discussion

This case report shows a 4-year follow up prospective evaluation of the virological and immunological patterns occurring in a primary CMV infection during pregnancy, resulting in severe cCMV. In this study the maternal CMV-specific cell-mediated immunity, a novel biomarker predictive of cCMV, was included along with the established widely used serological biomarkers (IgM/IgG/IgG avidity tests). This study showed that early after primary CMV infection, a robust and long-lasting CMV-specific cell-mediated immune response mounted and persisted with high levels up to 4 years after primary infection. The sustained antiviral immune response persisted years after viral clearance in blood and urine. This finding is in consistent with a prior study showing that Th1/Th2 immune imbalance may be the ultimate cause of cCMV [8,12]. Interesting findings emerged from the histopathological examination: at the placental level an abundant lymphocytic and plasma cellular infiltrate of villous stroma was concomitant with active viral replication and fetal dissemination. We speculate that this inflammatory condition caused by the viral replication along with the robust antiviral cell-mediated immune response facilitated the CMV transmission in utero, suggesting that an imbalanced Th1/Th2 immune response may be responsible for cCMV. The CMV specific cell-mediated immune response has been extensively studied in the solid organ and hematopoietic stem cell transplantation setting [13,14,15,16], suggesting that the antiviral cell-mediated immune reconstitution plays a positive role in limiting the CMV infection rate. Paradoxically, the cell-mediated immune response, known to be beneficial in transplant settings,. it seems to be a novel biomarker associated with an increased rate of cCMV in pregnant women. We speculate that at the local placental level, the inflammatory milieu provoked by the viral replication combined with an altered cytokine milieu caused by an unbalanced Th1/Th2 immune response may favor viral transmission and in utero dissemination. The high proportion of infiltrating immune cells may also provoke hypoxia-associated placental alterations resulting in fetal injury and neurological abnormalities.

## Figures and Tables

**Figure 1 viruses-15-00112-f001:**
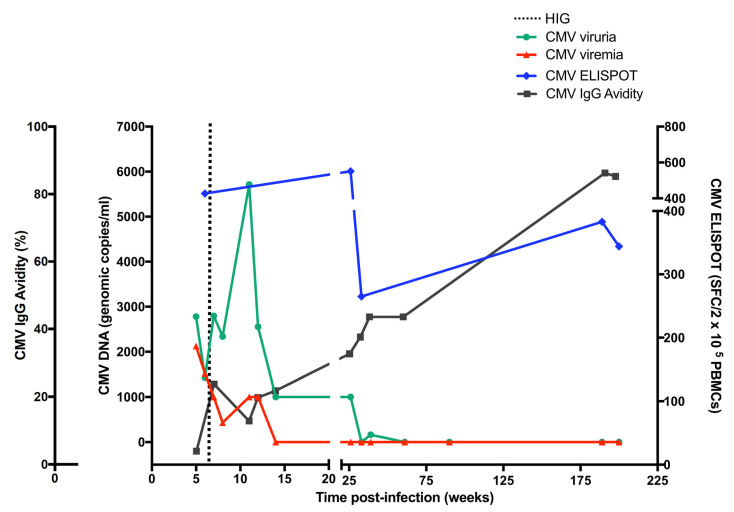
Prospective follow-up pattern of the CMV specific T-cell mediated-immunity (CMV-EliSpot), CMV IgG avidity, CMV viremia and viruria in a pregnant woman experiencing primary CMV infection.

**Figure 2 viruses-15-00112-f002:**
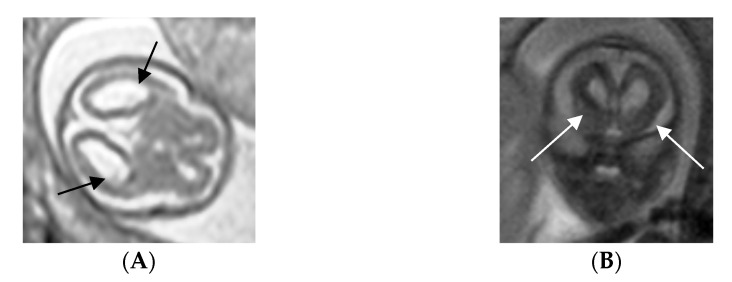
MRI section of fetal brain with arrows displaying lateral ventricular enlargement (**A**) and increased thickness of the germinal matrix (**B**) at 15 weeks of gestation.

**Figure 3 viruses-15-00112-f003:**
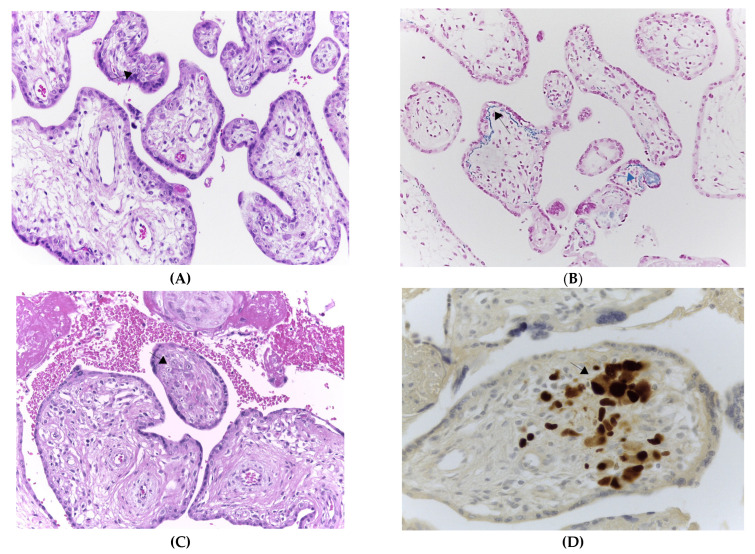
Placenta histological examination with arrows displaying: (**A**) Increased stromal fibrosis, decreased number of capillaries and presence of lymphocytes and plasma-cells (H&E stain, 200× magnification). (**B**) Hemosiderin depositions on trophoblastic basal membrane (Perls staining positive, 200× magnification). (**C**) Cytomegalic stromal cells with eosinophil intranuclear inclusion bodies (H&E stain, 200× and 400× magnification). (**D**) Immunostaining for CMV immediate early (IE1/2) antigens displaying a strong and diffuse nuclear positivity (400× magnification).

## Data Availability

Written informed consent has been obtained from the patient to publish this paper.

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
