# Peer review of "Four-Year Follow-Up of the Maternal Immunological, Virological and Clinical Settings of a 36-Year-Old Woman Experiencing Primary Cytomegalovirus Infection Leading to Intrauterine Infection"

_viruses, 2022, doi:10.3390/v15010112_

Round 1

Reviewer 1 Report

To my knowledge a systematic patient follow-up has not been published so far. The authors present an appropriate description of the affected child with cCMV as well as a long time follow up of the mothers immune responses. The authors draw their conclusions to an appropriate level for a case report. Question has been answered within the review form.

In summary, I do not feel that a more extensive report would help to improve the manuscript, since the authors limit themselves to a case report in an appropriate manner.

This is a nice little paper and merits publication.

Author Response

we thank the reviewer's comments and appreciate his/her efforts in revising our manuscript.

Reviewer 2 Report

This is a case report describing the immunological, clinical, and virological settings of a CMV-infected pregnant woman experiencing intrauterine CMV transmission that deliberately interrupted the pregnancy at 16 weeks of gestation.

The topic is relevant and new case reports are deserved to improve the knowledge about HCMV-related congenital malformations. The manuscript is well-written and organized. The figures are adequate and the analysis looks right.

An interesting issue that would improve the report is to include more details about the growth properties of the clinical strain isolated from saliva, compared to the widely used HCMV laboratory strains.

Moreover, it would be useful to set up a table summarizing the more relevant clinical parameters related to the mother and the fetus.

Below are some other minor improvements to improve the quality of the work:

-          Abstract, line 27, point 3: something is wrong in the sentence, please check

-          Line 42: a dot is missing

-          Line 43: “a non-invasive procedure…”, please specify which procedure you are referring to

-          Line 46-47: the link between the previous publication you cited and the current work is not immediately understandable; in my opinion, it should be better specified

-          Line 58: a dot is missing

-          Line 61: a dot is missing; specify the protocol number of the approval from the Ethical committee

-          Line 64-66: revise and make more fluent the sentence

-          Line 70: a verb is missing

-          Line 65-66: specify the HCMV viral load in the saliva

-          Figure 3D: specify the CMV antigen employed for immunostaining

-    References could be more updated and more varied in the choice of authors (almost 10 out of 28 references are self-citations)

Author Response

This is a case report describing the immunological, clinical, and virological settings of a CMV-infected pregnant woman experiencing intrauterine CMV transmission that deliberately interrupted the pregnancy at 16 weeks of gestation.

The topic is relevant and new case reports are deserved to improve the knowledge about HCMV-related congenital malformations. The manuscript is well-written and organized. The figures are adequate and the analysis looks right.

Authors reply: the authors thank and appreciate the reviewer comments.

An interesting issue that would improve the report is to include more details about the growth properties of the clinical strain isolated from saliva, compared to the widely used HCMV laboratory strains.

Authors reply: unfortunately in this circumstance the virus has not been isolated in vitro. The following statement has been added in the M&Ms for clarification: "The virus was not isolated from the saliva swabs."

Moreover, it would be useful to set up a table summarizing the more relevant clinical parameters related to the mother and the fetus.

Authors reply: since the manuscript is a short case report, we feel redundant to include a table with a summary of the data. No changes were made.

Below are some other minor improvements to improve the quality of the work:

-          Abstract, line 27, point 3: something is wrong in the sentence, please check

Authors reply: The authors corrected the sentence as follows " The fetal histological and pathological evidences suggest that a potent pro-inflammatory condition at the placental level that may lead to cCMV."

-          Line 42: a dot is missing

Authors reply: the authors thank the reviewer comment. The typing error has been corrected.

-          Line 43: “a non-invasive procedure…”, please specify which procedure you are referring to

Authors reply: the assay we refer to is performed on peripheral blood mononuclear cells. The proper reference is present in the manuscript however for clarification purposes the authors modified the statement as follows " In our prior study, a non-invasive procedure, the CMV EliSpot, performed on maternal peripheral blood isolated cells, revealed that elevated levels of cell-mediated immunity (CMI) specific for CMV, significantly correlated with higher risk of fetal transmission, thus suggesting that an imbalanced or altered pattern of Th1/Th2 response may be the ultimate cause of cCMV [8]."

-          Line 46-47: the link between the previous publication you cited and the current work is not immediately understandable; in my opinion, it should be better specified

Authors reply: we think that the re-written statement reported above addresses all the reviewer concerns.

-          Line 58: a dot is missing

Authors reply: the authors thank the reviewer comment. The typing error has been corrected.

-          Line 61: a dot is missing; specify the protocol number of the approval from the Ethical committee

Authors reply: the authors thank the reviewer comment. The typing error has been corrected. The Ethical committee data were transmitted to the journal along with protocol number and approvals.

-          Line 64-66: revise and make more fluent the sentence

Authors reply: the whole technical procedure was removed and referred to the citation. Now the sentence is " CMV-EliSpot data are sown as number of spots forming colonies (SFC) /2x105 PBMCs) [10].

-          Line 70: a verb is missing

Authors reply: the authors thank the reviewer comment. The error has been corrected and the technical procedures were removed and cited. The edited version reports "CMV-EliSpot data are sown as number of spots forming colonies (SFC) /2x105 PBMCs) [10]."

-          Line 65-66: specify the HCMV viral load in the saliva

Authors reply: there is no approved HCMV quantitative method for saliva. HCMV quantitative PCR standardization is approved for blood and amniotic fluid. No changes were made in the text.

-          Figure 3D: specify the CMV antigen employed for immunostaining

 Authors reply: the authors want to thank the reviewer for the point raised. The antigen staining were HCMV immediate early (IE1/2) proteins. The text has been changed as follows " Immunostaining for CMV immediate early (IE1/2) antigens displaying a strong and diffuse nuclear positivity"

-    References could be more updated and more varied in the choice of authors (almost 10 out of 28 references are self-citations)

Authors reply: according to the editor's suggestions, the references were modified and integrated including the most recent ones in the filed. The authors would like to keep the self-citations since these represent important information for the readers.